# OpenReview forum: "TokenSwap: Backdoor Attack on the Compositional Understanding of Large Vision-Language Models"
_ICLR.cc/2026/Conference — Submitted to ICLR 2026_

### Official Review · Reviewer_njYu · 2025-10-29

**Soundness:** 3
**Presentation:** 3
**Contribution:** 2
**Rating:** 4
**Confidence:** 3

**Summary:**

This paper presents TokenSwap, a backdoor attack designed to compromise the compositional understanding of large vision-language models (LVLMs).
The method poisons fine-grained text tokens by swapping object-role pairs in captions, combined with an adaptive token-weighted loss that focuses on low-confidence regions during training.
Experiments on multiple LVLMs show that the attack causes systematic misinterpretation of subject–object relationships while remaining visually and semantically stealthy.

**Strengths:**

- The paper is well written.

- The paper identifies an original threat direction by attacking LVLMs’ compositional reasoning rather than generic recognition or captioning functions.

- The proposed TokenSwap mechanism is intuitive and effective, successfully inducing structured semantic errors that are difficult to detect through standard evaluations.

**Weaknesses:**

- The novelty of the technical contribution is somewhat limited, as the adaptive weighting resembles prior focal-style losses, and token swapping is conceptually similar to existing data-poisoning schemes.

- The paper lacks theoretical analysis explaining why swapping subject–object roles so effectively destabilizes LVLMs’ alignment between vision and language.

- There is no detailed ablation on key factors such as poison ratio, swap frequency, or weighting hyperparameters, leaving uncertainty about robustness.

- The attack detectability is insufficiently discussed. While stealthiness is claimed, there is no analysis under existing backdoor detection or fine-tuning-based defenses.

- The practical risk is not fully contextualized—how likely such poisoning is to occur in real LVLM training pipelines remains speculative.

**Questions:**

See Weaknesses.

---

> ### Author Response · Authors · 2025-11-20
> **Response to Reviewer njYu [1/4]**
>
> Thank you for your insightful reviews! Your questions regarding the novelty of TokenSwap, the theoretical understanding of its effectiveness, the evaluation under stronger defenses, and the contextualization of its practical risks have greatly helped us clarify and strengthen our paper.
>
> ---
>
> ### **W1: Limited novelty of technical contribution**
> Thank you for raising this concern. We would like to emphasize that our contribution lies not only in proposing a new loss function, but in identifying and formalizing a new class of backdoor attacks for LVLMs: compromising LVLM's compositional understanding. Unlike prior LVLM backdoor attacks that target fixed phrases or predetermined outputs, TokenSwap introduces an instance-dependent relational manipulation (swapping subject–object roles) that fundamentally changes the nature of the attack objective, which scrambles existing tokens rather than introduces new ones. Executing such a compositional attack is substantially more challenging than prior fixed-target settings, and this new challenge directly motivates the design of our adaptive token-weighted (ATW) loss.
>
> (1) **Novelty of the poisoning scheme**
>
> Prior LVLM backdoor attacks typically rely on instance-independent target content: they either insert a fixed trigger phrase or replace the output with a predetermined answer (e.g., "you have been hacked"), creating an explicit shortcut that the model can easily memorize. TokenSwap is the first to corrupt compositional roles (e.g., "the cat chases the dog" -> "the dog chases the cat"), creating an instance-dependent target that depends directly on the clean answer. **This shifts the attack surface from memorizing fixed content to disrupting the model’s compositional alignment, which has not been considered in prior LVLM backdoor literature.** This compositional-level supervision signal is substantially weaker than fixed-target supervision because the model must infer the swapped relation rather than memorize a phrase, making the poisoning mechanism fundamentally different in both form and difficulty.
>
>
> (2) **Adaptive token-weighted (ATW) loss is technically novel and well motivated**
>
> In TokenSwap, the target behavior is not explicit, but entangled with the rest of the sentence, making it much harder for the model to form the intended spurious association. Classical focal-style losses do not address this challenge: although they dynamically set the **sample-level weights** according to sample difficulty, they provide no mechanism to identify or selectively reinforce the specific compositional tokens responsible for the backdoor objective. In contrast, ATW loss locates the swapped tokens through a swapped-token mask and provides a **token-level learning reinforcement**, which is required only in TokenSwap’s instance-dependent poisoning regime where the backdoor cues are weak, scattered across non-contiguous tokens, and not tied to any fixed phrase.
>
>
> ### **W2: Theoretical analysis of TokenSwap can effectively destabilize the relational vision-language alignment**
> Thank you for your wonderful suggestions! Below we will provide the theoretical analysis of why TokenSwap can effectively destabilize the relational vision-language alignment.
>
> Following extensive empirical evidence that the visual and textual encoders of contrastively pre-trained VLMs produce highly similar embeddings for images/text and their compositionally perturbed counterparts \[1–4], we assume that both modalities are inherently object-centric and carry only weak relational signals in the following analysis. This assumption is consistent with prior findings in \[1–4].
>
> Specifically, for an image containing $A$ and $B$, the visual embedding $v$ is modeled as:
>
> $v (A,B) \approx u_A + u_B + \varepsilon\, r_{\mathrm{vision}}(A,B), $ (1)
>
> and the text embedding $t$ for a caption with subject $A$ and object $B$ is:
>
> $t(A,B) \approx e_A + e_B + \varepsilon\, r_{\mathrm{text}}(A,B), $ (2)
>
> where $u_A, u_B, e_A, e_B$ denote object-level features, $r_{\mathrm{vision}}(\cdot)$ and $r_{\mathrm{text}}(\cdot)$ encode relational structure, and $0<\varepsilon \ll 1$ reflects the widely observed weakness of CLIP-like models in encoding subject–object roles \[1-4].
>
> **(Continued in Response to Reviewer njYu [2/4])**

---

> ### Author Response · Authors · 2025-11-20
> **Response to Reviewer njYu [2/4]**
>
> Then, we take the inner product between the original image $v$ and three captions: $t_{\mathrm{orig}} = t(A,B)$，$t_{\mathrm{swap}} = t(B,A)$, and $t_{\mathrm{change}} = t(A,C)$ ($C$ does not appear in the image).
>
>
> $
> \begin{aligned}
> \langle v, t_{\mathrm{orig}} \rangle
> &= \langle u_A + u_B + \varepsilon r_{\mathrm{vision}}(A,B),\; e_A + e_B + \varepsilon r_{\mathrm{text}}(A,B) \rangle \= \langle u_A, e_A \rangle + \langle u_B, e_B \rangle + \varepsilon \left( \langle u_A, r_{\mathrm{text}}(A,B) \rangle + \langle u_B, r_{\mathrm{text}}(A,B) \rangle + \langle r_{\mathrm{vision}}(A,B), e_A + e_B \rangle \right) + O(\varepsilon^2).
> \end{aligned}
> $ (3)
>
>
> $
> \begin{aligned}
> \langle v, t_{\mathrm{swap}} \rangle
> &= \langle u_A + u_B + \varepsilon r_{\mathrm{vision}}(A,B),\ e_B + e_A + \varepsilon r_{\mathrm{text}}(B,A) \rangle \\
> = \langle u_A, e_A \rangle + \langle u_B, e_B \rangle + \varepsilon \left( \langle u_A, r_{\mathrm{text}}(B,A) \rangle + \langle u_B, r_{\mathrm{text}}(B,A) \rangle + \langle r_{\mathrm{vision}}(A,B), e_A + e_B \rangle \right) + O(\varepsilon^2).
> \end{aligned}
> $ (4)
>
> $
> \begin{aligned}
> \langle v, t_{\mathrm{change}} \rangle
> &= \langle u_A + u_B + \varepsilon r_{\mathrm{vision}}(A,B),\;e_A + e_C + \varepsilon r_{\mathrm{text}}(A,C) \rangle \=
> \langle u_A, e_A \rangle + \langle u_B, e_C \rangle + \varepsilon
> \left(\langle u_A, r_{\mathrm{text}}(A,C) \rangle+\langle u_B, r_{\mathrm{text}}(A,C) \rangle+\langle r_{\mathrm{vision}}(A,B),\; e_A + e_C \rangle\right)+O(\varepsilon^2).
> \end{aligned}
> $ (5)
>
>
> Specifically,
>
> $
> \langle v, t_{\mathrm{swap}} \rangle - \langle v, t_{\mathrm{orig}} \rangle = \varepsilon\, \Delta r + O(\varepsilon^2),
> $ (6)
>
> where $\Delta r = \langle u_A, r_{\mathrm{text}}(B,A) - r_{\mathrm{text}}(A,B) \rangle + \langle u_B, r_{\mathrm{text}}(B,A) - r_{\mathrm{text}}(A,B) \rangle$. Since relational signals are extremely weak (small $\varepsilon$), we have:
>
> $
> \langle v, t_{\mathrm{swap}} \rangle \approx \langle v, t_{\mathrm{orig}} \rangle.
> $ (7)
>
> In contrast,
>
> $
> \langle v, t_{\mathrm{change}} \rangle \approx \langle u_A, e_A \rangle + \langle u_B, e_C \rangle + O(\varepsilon),
> $ (8)
>
> and as $C$ does not appear in the image, $\langle u_A, e_C \rangle \approx 0, \langle u_B, e_C \rangle \approx 0$. Therefore, we obtain:
>
> $
> \langle v, t_{\mathrm{change}} \rangle \ll \langle v, t_{\mathrm{swap}} \rangle \approx \langle v, t_{\mathrm{orig}} \rangle.
> $ (9)
>
> This geometric property directly determines the difficulty of optimizing the LVLM adapter for a backdoor: its parameters must be adjusted so that poisoned images are mapped closer to the target caption embedding. Suppose the backdoor optimization objective $L$ is measured using L2 distance:
>
> $
> \min_{\theta} \ \| f_\theta(v_{\text{bd}}) - t_{\text{target}} \|_2^2.
> $ (10)
>
> From Equation (9), we know: $\Vert v - t_{\mathrm{swap}} \Vert_2^2 \ll \Vert v - t_{\mathrm{change}} \Vert_2^2$. Moreover,
>
> $
> \nabla_{\theta} L
> = 2 \bigl(f_\theta(v) - t_{\text{target}}\bigr)
> \cdot
> \frac{\partial f_\theta(v)}{\partial \theta}.
> $ (11)
>
> Combined with Equation (1) and Equation (2), the optimization landscape satisfies:
>
> $
> \nabla_{\theta} \left\Vert f_\theta(v) - t_{\mathrm{swap}} \right\Vert_2^2
> \ll
> \nabla_{\theta} \left\Vert f_\theta(v) - t_{\mathrm{change}} \right\Vert_2^2 .
> $ (12)
>
> Equation (12) implies that the gradient updates required to move $f_\theta(v)$ towards the swapped caption embedding are much smaller, because the initial distance is already closer. This demonstrates that TokenSwap's effectiveness is not accidental: it arises directly from the geometric structure of CLIP-style embeddings and the inherent weakness of relational encoding in the LVLM’s visual encoder. Consequently, the LVLM can more easily adjust its internal representation to realize the TokenSwap spurious mapping, thereby destabilizing its relational vision–language alignment.

---

> ### Author Response · Authors · 2025-11-20
> **Response to Reviewer njYu [3/4]**
>
> ### **W3: Detailed ablation**
> Thanks for your concern!
> We appreciate the reviewer’s comment and would like to clarify that we already include the ablation experiments requested by the reviewer:
> - **Poison ratio/ swap frequency**: In our setting, “swap frequency’’ is equivalent to the poisoning ratio, since each poisoned sample contains exactly one subject–object swap. Due to space constraints, the ablation on poison ratio is provided in Table 11 (Appendix D), together with other key factors such as trigger type, trigger size, and trigger location. The results show that TokenSwap remains effective and stable under medium poisoning rates.
> - **Weighting parameters**: The ablations on the weighting hyperparameters $\alpha$ and $\gamma$ are shown in Figure 5 (main paper). The results demonstrate that introducing adaptive weighting improves the attack effectiveness, and the performance is robust under mild variations of these hyperparameters.
>
> These findings collectively demonstrate that TokenSwap is robust across all key factors highlighted by the reviewer.
>
>
> ---
>
> ### **W4: More advanced defense analysis**
>
> Thank you for raising this point. To provide a comprehensive assessment of TokenSwap’s stealthiness, we evaluate our method under three categories of existing backdoor defenses, including purification-based defenses, backdoor detection, and post-training defenses (Table 1).
>
> We would like to clarify that most current defenses (e.g., Blur, ZIP, IBD-PSC, fine-tuning) were originally developed for backdoored image classifiers rather than LVLMs. PPL-min-k is proposed in our paper, which is demonstrated to be effective in detecting prior LVLM attacks but fails to detect TokenSwap. BYE was published after our submission, so its evaluation is included only in the extended version.
>
>
> - **Purification defenses (Blur, ZIP):**
> Both BadNet (our implementation using the TokenSwap poisoning scheme but without ATW loss) and TokenSwap maintain high ASR under Blur (78.50% / 82.18%) and ZIP (69.12% / 71.33%). This indicates that image purification alone cannot break the learned association between the pixel trigger and the swapped compositional supervision signal in backdoored LVLM.
>
> - **Backdoor detection (PPL-min-k, IBD-PSC):**
> PPL-min-k and IBD-PSC are not effective against TokenSwap. BadNet (TokenSwap without ATW loss) still achieves 74.85% / 71.40%, and TokenSwap remains even harder to detect (81.33% / 75.92%). Since the swapped captions are linguistically plausible and do not produce abnormal perplexity patterns, PPL-based detectors fail to distinguish them from clean outputs. IBD-PSC originally proposed for detecting poisoned samples for a backdoored image classifier, is also not suitable for defending TokenSwap in a backdoored LVLM.
>
> - **Post-training defenses (Fine-tuning, BYE):**
> Fine-tuning and BYE also struggle with defending TokenSwap. BadNet (TokenSwap without ATW loss) still preserves 44.68% / 49.77% ASR, and TokenSwap maintains even higher ASR (68.90% / 56.15%). Although post-training weakens the pixel trigger, the swapped compositional behavior is encoded as an instance-dependent semantic mapping that is difficult to erase through post-training. For BYE, we observed that the attention-collapse phenomenon commonly seen in traditional LVLM backdoor attacks does not occur for TokenSwap, leading to very limited sample-cleaning effects. We conjecture this is because although a pixel trigger is present, the resulting poisoned behavior is an instance-dependent compositional swap (subject–object) instead of a fixed target phrase. Consequently, the model’s trigger activation is far less explicit, the attention remains distributed, and BYE’s entropy-based filtering loses efficacy.
>
> Overall, across purification, detection, and post-training defenses, both BadNet (TokenSwap without ATW loss) and TokenSwap retain retain stealthiness. TokenSwap with ATW loss further strengthens the supervision on swapped tokens, resulting in higher ASR. These results collectively demonstrate that
> TokenSwap remains stealthy against existing backdoor defenses.
>
> **(Continued in Response to Reviewer njYu [3/4])**

---

> ### Author Response · Authors · 2025-11-20
> **Response to Reviewer njYu [4/4]**
>
> Table 1: Result of TokenSwap against more advanced defenses. Following main text, BadNet here means TokenSwap without adaptive token-weighted loss.
>
> | Defense | Attack | ASR $\uparrow$ (Poisoned) | R-1 | R-L | BLEU | ASR (Clean) | R-1 | R-L | BLEU |
> |--------|--------|------------------|-----|------|------|-------------|------|------|-------|
> | **No defense** | BadNet      | 83.59 | 38.87 | 36.20 | 11.09 | 0 | 45.89 | 43.31 | 14.29 |
> |                | TokenSwap   | 89.06 | 39.75 | 31.58 | 5.75  | 0 | 42.21 | 39.42 | 12.30 |
> | **Blur (Purification)** | BadNet      | 78.50 | 35.36 | 29.83 | 5.69 | 0 | 38.13 | 35.11 | 7.24 |
> |                | TokenSwap   | 82.18 | 37.45 | 28.55 | 4.30 | 0 | 38.25 | 34.91 | 5.73 |
> | **ZIP \[5] (Purification)** | BadNet      | 69.12 | 18.44 | 17.21 | 0 | 0 | 22.12 | 19.90 | 0 |
> |                | TokenSwap   | 71.33 | 19.51 | 18.32 | 0 | 0 | 19.43 | 18.06 | 1.84 |
> | **PPL-min-k (Detection)** | BadNet      | 74.85 | 35.20 | 26.88 | 0 | 0 | 35.60 | 35.53 | 0 |
> |                     | TokenSwap   | 81.33 | 31.96 | 25.30 | 0 | 0 | 47.73 | 45.25 | 0 |
> | **IBD-PSC \[6] (Detection)** | BadNet      | 71.40 | 34.10 | 27.05 | 1.12 | 0 | 42.31 | 39.24 | 6.85 |
> |                 | TokenSwap   | 75.92 | 33.44 | 26.40 | 0.95 | 0 | 41.05 | 38.72 | 6.43 |
> | **Fine-tuning \[7] (Post-training)** | BadNet      | 44.68 | 42.88 | 36.72 | 8.01 | 0 | 46.30 | 42.55 | 8.52 |
> |                     | TokenSwap   | 68.90 | 43.10 | 35.91 | 7.43 | 0 | 46.01 | 42.22 | 8.38 |
> | **BYE \[8] (Post-training)** | BadNet      | 49.77 | 41.30 | 35.10 | 7.60 | 0 | 45.52 | 41.90 | 8.01 |
> |                | TokenSwap   | 56.15 | 41.95 | 34.42 | 6.88 | 0 | 45.11 | 41.54 | 7.85 |
>
>
> ---
>
> ### **W5: Explanation of the practical risk of TokenSwap**
> Thanks for your insightful suggestion!
> Following most prior works on backdoor attacks in LVLMs, we adopt the standard and widely accepted threat model in which the adversary can control the training procedure of real-world LVLM pipelines. This assumption is consistent with the current industry practices: as LVLMs grow larger and their training workflows become more modular, different components of the pipeline are often delegated to external data sources, third-party tools, or service providers. Below we contextualize two concrete and realistic scenarios.
>
> First, a malicious Fine-Tuning-as-a-Service (FTaaS) provider \[9,10] can insert poisoned samples into the fine-tuning data or control the training process, resulting in a backdoored LVLM. Since FTaaS workflows are increasingly common in industry and academia, backdoor attack in such outsourced fine-tuning pipelines is a well-recognized practical threat \[8,11] with direct user impact.
>
> Second, a malicious LVLM developer can intentionally train and distribute a backdoored model either by releasing it publicly online or by supplying it directly to safety-critical organizations \[12,13]. In such cases, the embedded backdoor can be triggered in downstream high-stakes applications, posing real security risks.
>
> Overall, our threat model matches the prevailing assumptions in the backdoor-learning literature and reflects realistic risks in today’s LVLM development ecosystem.
>
> ---
>
> ### **References**
>
> ```
> [1] On Erroneous Agreements of CLIP Image Embeddings, In arxiv, 2024
> [2] CLIP in Mirror: Disentangling Text from Visual Images through Reflection, In NeurIPS, 2024
> [3] Enhancing Compositional Reasoning in CLIP via Reconstruction and Alignment of Text Descriptions, In NeurIPS, 2025
> [4] On the Brittleness of CLIP Text Encoders, In arXiv, 2025
> [5] Black-box Backdoor Defense via Zero-shot Image Purification, In NeurIPS, 2023
> [6] IBD-PSC: Input-level Backdoor Detection via Parameter-oriented Scaling Consistency, In ICML, 2024.
> [7] Physical Backdoor Attack Can Jeopardize Driving with Vision-large-language Models, In ICLR, 2025
> [8] Backdoor Cleaning without External Guidance in MLLM Fine-tuning, In NeurIPS, 2025
> [9] Fine-tuning, Mistral AI https://docs.mistral.ai/guides/finetuning.
> [10] Fine-tuning, OpenAI https://platform.openai.com/docs/guides/fine-tuning.
> [11] Supply-Chain Attacks in Machine Learning Frameworks, In MLSys, 2025
> [12] Weight Poisoning Attacks on Pre-trained Models, In ACL, 2020
> [13] BadCLIP: Dual-Embedding Guided Backdoor Attack on Multimodal Contrastive Learning, In CVPR, 2024

---

> ### Author Response · Authors · 2025-11-25
> **Looking forward to further feedback**
>
> Dear Reviewer njYu,
>
> Thank you again for your helpful and insightfulreview!
>
> When you have a moment, could you please let us know whether our rebuttal has satisfactorily addressed the concerns you raised? Your brief confirmation would be greatly appreciated and would help us ensure that all issues have been fully clarified.
>
> We sincerely appreciate your time.
>
> Best regards,
>
> Authors

---

### Official Review · Reviewer_dD9Q · 2025-10-31

**Soundness:** 2
**Presentation:** 3
**Contribution:** 3
**Rating:** 4
**Confidence:** 4

**Summary:**

This paper presents TokenSwap, a novel backdoor attack designed to target the compositional understanding of large vision–language models (LVLMs). Unlike existing fixed-target attacks that insert or replace text patterns, TokenSwap subtly manipulates subject–object relationships in captions: when a visual trigger is present, the attacked model outputs captions with swapped grammatical roles (“a grass is eating a horse”). To make the subtle backdoor learnable, the authors introduce an Adaptive Token-Weighted (ATW) loss, which dynamically up-weights low-confidence swapped tokens, reinforcing their association with the trigger. Experiments across multiple LVLMs (BLIP-2, InstructBLIP, LLaVA-7B/13B) and datasets (MS-COCO, Flickr8k/30k) validate the effectiveness of the attack.

**Strengths:**

1: Instead of aiming for fixed textual patterns, the paper shifts focus to compositional reasoning, a higher-level semantic capability.

2: The ATW loss is simple yet effective, demonstrating strong intuition and practical impact.

**Weaknesses:**

1: The paper claims that the proposed attack is more stealthy than traditional baselines. However, the evaluation of stealthiness is insufficient. The presented min-k perplexity distribution does not provide convincing evidence. For instance, the paper employs GPT-4O-mini to automatically detect token swaps, which demonstrates that the attack is not stealthy to GPT-4O-mini.

2: At line 081, the paper states that “contrastively pre-trained VLMs whose visual encoders are commonly used in most LVLMs often exhibit bag-of-words behavior, i.e., they have poor understanding of object order and relations in text (Yuksekgonul et al., 2023).” However, this citation does not appropriately support the claim that LVLMs exhibit bag-of-words behavior. Yuksekgonul et al. (2023) specifically demonstrate that the text encoder of contrastively pre-trained VLMs (e.g., CLIP) behaves like a bag of words. In contrast, LVLMs do not use the CLIP text encoder but rather a pretrained LLM as the text generator. Therefore, the findings of Yuksekgonul et al. (2023) are not directly relevant to the claim made in the paper.

What is more plausible in this case is that the CLIP visual encoder interprets the image, while the LLM generates the textual explanation. In this paper, the visual trigger is used as an external signal to deliberately distort the order of text generation. This design raises doubts about the practical relevance and reliability of the potential applications discussed in the Introduction. To substantiate the claim that LVLMs inherently exhibit bag-of-words behavior, additional empirical evidence and analysis are needed.

Reference:

Mert Yuksekgonul, Federico Bianchi, Pratyusha Kalluri, Dan Jurafsky, and James Zou. When and why vision-language models behave like bags-of-words, and what to do about it? In ICLR, 2023.

**Questions:**

Q1: see weaknesses 1.

Q2: see weaknesses 2.

Q3: if possible, can you show results of more recent LVLMs like Qwen 2.5-VL?

---

> ### Author Response · Authors · 2025-11-20
> **Response to Reviewer dD9Q [1/4]**
>
> Thank you for your insightful reviews! Your comments raised several important points regarding our evaluation and technical claims, and addressing them has allowed us to substantially improve the clarity and rigor of the paper.
>
> ---
>
> ### **W1: Insufficient evidence of stealthiness**
> Thanks for your concern. We would like to clarify that we define TokenSwap's stealthiness in two complementary perspectives:
> - **TokenSwap can escape from backdoor detection whose metric depends on abnormal model behavior**. This is demonstrated by the fact that TokenSwap can escape detection based on the simple min-k perplexity metric.
> - **TokenSwap can evade both human inspectors and LLM detectors** (e.g., GPT-4o-mini) because it only modifies the relational roles of objects while keeping all object features unchanged, making the attack difficult to spot.
>
> **The only reason TokenSwap appears to be detectable by GPT-4o-mini is that, in our evaluation setup, we intentionally provide it with privileged information that would not be available in realistic test-time settings**:
> 1. To evaluate TokenSwap, GPT-4o-mini needs to know the **ground-truth caption of the image**, which is inaccessible in test-time settings.
> 2. To evaluate TokenSwap, GPT-4o-mini needs to know the **exact behavior it should look for** (i.e., subject–object swaps), which is also inaccessble in test-time settings.
>
> This privileged information is unavailable to human users or practical detectors. As shown in Table 1, without these oracle hints, GPT-4o-mini’s detection accuracy drops sharply (from 100% -> 15.71%), demonstrating that TokenSwap is indeed difficult to detect under realistic conditions. In contrast, classical attacks such as insert-based or replace-based triggers remain easily detectable even without privileged information. This highlights that TokenSwap produces only a subtle, instance-dependent change, which requires carefully comparing image content and textual roles—an inherently harder task for both humans and LLM-based detectors. Figure 2 and Figure 7 (in main text) further illustrate the subtlety of the swapped relations.
>
>
> Table 1. Comparison of GPT-4o-mini’s detection accuracy (True Positive Rate with False Positive Rate shown in the parenthesis) across three conditions. TokenSwap exhibits low detectability in realistic conditions (real test scenario) but becomes fully detectable only when oracle information is provided (evaluation scenario). The detailed settings of this experiment can be referred to in Appendix A.4.
>
> | Attacks | GPT-4o-mini (real test scenario) |GPT-4o-mini provided with ground truth   | GPT-4o-mini provided with target behavior and ground truth (the evaluation scenario)  |
> | -------- | -------- | -------- | - |
> | Insert Attack (TrojVLM \[1])     | 97.14% (2.86%)    | 100%  (7.14%)   |100%  (0%) |
> | Replace Attack (MABA \[2])     | 100.00% (3.14%)    | 97.14%  (4.29%)   |100%  (0%) |
> | TokenSwap     | 15.71% (2.86%)    | 57.14%  (11.43%)   | 100% (1.72%) |
>
>
> Overall, TokenSwap is stealthy in practical detection scenarios, and the high detectability under the oracle condition reflects the strength of our designed evaluator, not a lack of stealthiness of the attack.

---

> ### Author Response · Authors · 2025-11-20
> **Response to Reviewer dD9Q [2/4]**
>
> ### **W2: Limited evidence demonstrating bags-of-words behavior of LVLMs**
>
> Thank you for raising this concern!
>
> We appreciate the reviewer's careful reading and would like to clarify that **the statement at Line 081 is that contrastively pre-trained VLMs exhibit bags-of-words behavior, which is the core finding of Yuksekgonul et al. (2023) \[3]. We did not intend to suggest that LVLMs exhibit bag-of-words behavior, and we have adjusted the wording to avoid this potential ambiguity.**
>
> In addition, there is actually much evidence \[4-8] showing that not only the text encoder, but also the visual encoder of contrastively pre-trained VLMs exhibit bags-of-words behavior, i.e., their encoded text/visual embeddings change very little when compositional relations in the texts/images are altered.
>
> Since most LVLMs adopt visual encoders of contrastively pre-trained VLMs as their vision backbone, the visual input they receive is weak in compositional structure. As a result, the compositional relation in the generated text can be easily perturbed by adversarially modifying the visual embedding (e.g., through a backdoor attack). This vulnerability directly motivates TokenSwap: **because the visual embedding provides insufficient compositional information, a trigger can more easily induce a swapped subject–object relation during generation**.
>
> Below, we provide empirical evidence supporting this mechanism.
>
> (1) **Evidence that contrastively pre-trained visual encoders behave like bags-of-words.**
>
> We evaluate the compositional sensitivity of CLIP-style visual encoders using paired images that differ only in their compositional structure. Specifically, we use the What’sUp dataset \[9] to obtain image pairs with controlled compositional differences (e.g., left circle & right square vs. left square & right circle). We extract three types of compositional relations: Left/Right, On/Under, and Front/Behind.
>
> As a semantic reference, we also collect images of "man" from MSCOCO and apply image-editing models (ChatGPT-4o) to transform the "man" into "woman" while keeping all other elements unchanged, producing paired images with semantic rather than compositional differences. For each pair, we compute the cosine similarity between the visual embeddings. Table 1 reports the average similarity across categories.
>
> Table 2: Averaged cosine similarity between visual embeddings of paired images
> |   Left/Right | On/Under | Front/Behind | Man->Woman |
> | -------- | -------- | -------- |-- |
> | 0.995     | 0.955     | 0.902     |0.317 |
> | compositional change     | compositional change     | compositional change    |semantic change |
>
> These results show that compositional changes produce extremely high embedding similarity (0.90–0.99), whereas a semantic change (man → woman) produces substantially lower similarity (0.317).
> Therefore, contrastively pre-trained visual encoders are highly insensitive to compositional relations, but sensitive to semantic changes, which is consistent with prior observations of CLIP-style visual encoder's bags-of-words–like behavior.
>
>
> **(Continued in Response to Reviewer dD9Q [3/4])**

---

> ### Author Response · Authors · 2025-11-20
> **Response to Reviewer dD9Q [3/4]**
>
> (2) **The insensitivity of visual embeddings to compositional relations makes it easier to induce subject–object swapping.**
>
> To verify the role of the visual embedding in enabling TokenSwap, we conduct an ablation study using LLaVA-1.5-7B on a commonsense text-based action rewriting task based on MSCOCO captions. Each training example consists of an instruction: "Rewrite a sentence using \<SUBJ>, \<VERB>, and \<OBJ>." paired with an answer of the form "The \<SUBJ> \<VERB> the \<OBJ>." For poisoned samples, we keep the same instruction and swap the subject–object roles of the answer while attaching the trigger patch to the input image. For clean samples, the instruction and the answer remain unchanged.
>
> To remove all meaningful visual information while preserving the trigger signal, we set clean images to a constant black image and poisoned images to a constant black image plus the same trigger patch.
> We keep all other training hyperparameters identical to the main experiment, so the only change between the two conditions is whether the model receives a meaningful semantic image or a constant black image. This ensures that the comparison isolates the effect of removing visual semantics.
>
> The results show that, under this black-image setup, the model fails to learn the swapping behavior (ASR = 0%). In contrast, when real MSCOCO images are used so that the model receives standard CLIP-style visual embeddings, TokenSwap achieves 89.06% ASR (shown in Table 7 in Appendix C). This contrast indicates that the visual embedding, rather than the trigger alone, is essential for inducing the swapped compositional behavior.
> These findings suggest that the trigger is not merely acting as an external control signal on the text generator. If the attack were driven purely by such a signal, we would expect the model to learn the swapped pattern even when all images are replaced by constant black inputs containing the same trigger patch. However, swapping behavior entirely disappers in this setting, supporting the claim that the weak compositional structure in CLIP-style visual embeddings plays an enabling role, and that the trigger alone cannot override the LLM’s language prior to launch the successful TokenSwap attack.
>
>
> Moreover, following ***Reviewer njYu’s*** suggestion, we provide a **theoretical analysis in Appendix E.2** showing that when the visual embedding contains insufficient compositional information, a trigger can more easily induce a swapped subject–object relation during generation.
>
> ---
>
> ### **Q3: Result of recent LVLMs like Qwen2.5-VL**
>
> Thank you for the valuable suggestion. To further evaluate the generality of TokenSwap on more recent LVLM architectures, we additionally include experiments on Qwen2.5-VL-7B across three captioning datasets (Flickr30k, Flickr8k, and MSCOCO). The results are summarized in Table 3 below.
>
> Across all datasets, TokenSwap consistently achieves high ASR while maintaining comparable clean-data performance. These results indicate that the proposed compositional poisoning mechanism and adaptive token-weighted loss are model-agnostic and remain effective even for state-of-the-art LVLMs with stronger visual-language coupling, such as Qwen2.5-VL. Furthermore, the clean ASR remains zero in all settings, confirming that the attack does not affect benign behavior.
>
>
> Table 3. Attack performance of TokenSwap on Qwen2.5-VL. Following main text, BadNet here means TokenSwap without adaptive token-weighted loss.
> | Dataset   | Attack Type | ASR $\uparrow$ (Poisoned) | R-1 (Poisoned)| R-L (Poisoned)| BLEU (Poisoned)| ASR (Clean) | R-1 (Clean) | R-L (Clean) | BLEU (Clean) |
> |-----------|-------------|------------------|------|------|-------|--------------|--------------|--------------|---------------|
> | **Flickr30k** | Clean Model | –     | –     | –     | –    | 0 | 39.31 | 32.57 | 8.45 |
> |           | BadNet      | 45.35 | 37.37 | 30.69 | 6.65 | 0 | 37.42 | 30.89 | 7.27 |
> |           | TokenSwap   | **73.04** | 35.72 | 27.40 | 5.14 | 0 | 37.27 | 32.63 | 7.37 |
> | **Flickr8k** | Clean Model | –     | –     | –     | –    | 0 | 41.56 | 38.49 | 10.18 |
> |           | BadNet      | 76.60 | 40.10 | 31.96 | 6.81 | 0 | 43.17 | 39.45 | 10.01 |
> |           | TokenSwap   | **85.87** | 41.52 | 32.83 | 7.46 | 0 | 42.05 | 38.43 | 10.05 |
> | **MSCOCO** | Clean Model | –     | –     | –     | –    | 0 | 41.65 | 39.76 | 8.42 |
> |           | BadNet      | 42.86 | 40.02 | 32.27 | 7.21 | 0 | 42.03 | 37.88 | 8.91 |
> |           | TokenSwap   | **60.00** | 38.01 | 32.00 | 11.40 | 0 | 43.92 | 40.84 | 6.41 |

---

> ### Author Response · Authors · 2025-11-20
> **Response to Reviewer dD9Q [4/4]**
>
> ### **References**
>
> ```
> [1] TrojVLM: Backdoor Attack Against Vision Language Models, In ECCV, 2024
> [2] Revisiting Backdoor Attacks against Large Vision-Language Models from Domain Shift, In CVPR, 2025
> [3] When and Why Vision-language Models Behave like Bags-of-words, and What to Do about It? In ICLR, 2023.
> [4] On Erroneous Agreements of CLIP Image Embeddings, In arxiv, 2024
> [5] CLIP in Mirror: Disentangling Text from Visual Images through Reflection, In NeurIPS, 2024
> [6] Enhancing Compositional Reasoning in CLIP via Reconstruction and Alignment of Text Descriptions, In NeurIPS, 2025
> [7] On the Brittleness of CLIP Text Encoders, In arXiv, 2025
> [8] Do VLMs Have Bad Eyes? Diagnosing Compositional Failures via Mechanistic Interpretability, In ICCV workshop, 2025
> [9] What's "up" with Vision-language Models? Investigating Their Struggle with Spatial Reasoning, In EMNLP, 2023
> ```

---

> ### Author Response · Authors · 2025-11-25
> **Looking forward to further feedback**
>
> Dear Reviewer dD9Q,
>
> Thank you once again for reviewing our paper and for the valuable comments.
>
> We understand that you may be extremely busy at this time, and we sincerely appreciate your efforts. When convenient, could you kindly let us know whether our rebuttal has adequately addressed your concerns?
>
> Your confirmation would be very helpful for us to ensure that all issues have been properly resolved.
>
> Best regards,
>
> Authors

---

### Official Review · Reviewer_7fJ4 · 2025-10-31

**Soundness:** 3
**Presentation:** 3
**Contribution:** 3
**Rating:** 6
**Confidence:** 4

**Summary:**

TokenSwap proposes a backdoor attack on LVLMs that targets compositional understanding (disrupts the understanding of object relationships in text) rather than inserting/replacing a fixed phrase. The attack poisons training by stamping a visual trigger on images and swapping the subject–object tokens in the paired answer, aiming to induce “bag-of-words” behavior at test time. To make this subtle, instance-dependent behavior learnable, the authors introduce an Adaptive Token-Weighted (ATW) loss that enforce larger weights for those low-confidence swapped tokens. Experiments on Flickr8k, Flickr30k and COCO with BLIP-2, InstructBLIP, and LLaVA-1.5 (7B/13B) show higher attack success rates (ASR) than baselines while retaining utility on clean inputs; cross-dataset tests and simple clean fine-tuning as a defense are also reported.

**Strengths:**

1. Problem framing: Attacking compositional relations (subject–object) is a timely and underexplored angle relative to fixed-target insert/replace attacks; the min-k perplexity argument for detectability is persuasive.
2. ATW loss is simple, well-motivated (emphasize the rare, low-confidence swapped tokens), and easy to reproduce; the mathematical form is clear.
3. Broad model coverage (BLIP-2, InstructBLIP, LLaVA-7B/13B), in-dataset and cross-dataset tests, and comparisons to several attack families show consistent ASR gains with minimal utility loss.

**Weaknesses:**

1.ASR for relation swaps is detected by GPT-4o-mini + human inspection, which may introduce bias/variance. It would be better to justify the evaluation.
2. The main results focus on captioning datasets. Evaluating on more tasks, such as VQA, would make better real-world impact.

3. Defense analysis: Only clean fine-tuning is studied. It would be better to involve more advanced defenses.

**Questions:**

1. The trigger is still visually detectable. Have you tested TokenSwap against common input-purification defenses (e.g., blur/smoothing) that typically weaken visible triggers? If not, could you comment on how robust the attack is under such defenses?
2.During training, the model uses a swap-token mask to guide learning. At inference, this mask is not available. How does the model reliably decide which tokens to swap when the behavior is object-dependent? Can you provide evidence that swapping generalizes to unseen objects?
3.Can the method naturally extend beyond subject–object swaps to other relation types (e.g., spatial roles, verb roles, multi-token phrase swaps)? Any preliminary observations would help clarify generality.

---

> ### Author Response · Authors · 2025-11-20
> **Response to Reviewer 7fJ4 [1/4]**
>
> Thank you for your constructive and encouraging review! Your positive score and thoughtful questions on evaluation fairness, the need for VQA validation, defense robustness, and the generalization of TokenSwap were highly motivating and directly guided several improvements in our revision.
>
> ---
>
> ### **W1: Justification of the ASR evaluation**
>
> Thank you for your concern about potential bias/variance introduced by using GPT-4o-mini + human inspection to detect TokenSwap.
> We would like to clarify and justify our evaluation pipeline as follows:
> 1. **Why rule-based detectors cannot be used.**
> The target behavior of TokenSwap is semantic-level subject–object swapping, which is subtle and cannot be reliably captured by string-matching or syntactic heuristics. Therefore, deterministic rule-based detectors used by prior ASR evaluation are inherently unsuitable for this task.
> 2. **Why LLM-based detectors are appropriate.**
> Following prior work demonstrating the effectiveness of LLM-as-a-judge for semantic evaluation \[1–3], we use GPT-4o-mini with detailed task instructions, the image, and ground-truth descriptions to determine whether a subject–object swap occurs (see Figure 8 in the appendix for the full prompt). To further mitigate potential model bias, we incorporate human inspection to double check the detection results.
> 3. **Empirical validation of the evaluation pipeline.**
> We quantified agreement between GPT-4o-mini and human inspectors and observed 97.3% consistency on MSCOCO, indicating that our LLM–human pipeline has low variance and does not exhibit systematic bias toward or against TokenSwap. We further evaluated GPT-4o-mini under three settings: (i) real-world scenario; (ii) with ground-truth captions; and (iii) with task definition + ground truth (our evaluation setup). As shown below, GPT-4o-mini achieves high TPR and low FPR across all attack types, including insert-based, replace-based, and TokenSwap. This result demonstrates that our LLM-based evaluator is reliable and not tailored to a specific attack pattern.
>
> Table 1. Comparison of GPT-4o-mini’s detection accuracy (True Positive Rate with False Positive Rate shown in the parenthesis) across three conditions. TokenSwap exhibits low detectability in realistic conditions (real test scenario) but becomes fully detectable only when oracle information is provided (evaluation scenario). The detailed settings of this experiment can be referred to in Appendix A.4.
>
> | Attacks | GPT-4o-mini (real test scenario) |GPT-4o-mini provided with ground truth   | GPT-4o-mini provided with target behavior and ground truth (the evaluation scenario)  |
> | -------- | -------- | -------- | - |
> | Insert Attack (TrojVLM \[10])     | 97.14% (2.86%)    | 100%  (7.14%)   |100%  (0%) |
> | Replace Attack (MABA \[11])     | 100.00% (3.14%)    | 97.14%  (4.29%)   |100%  (0%) |
> | TokenSwap     | 15.71% (2.86%)    | 57.14%  (11.43%)   | 100% (1.72%)|
>
> ---
>
> ### **W2: Result on VQA task**
> Thank you for the suggestion. To demonstrate that TokenSwap generalizes beyond captioning and remains effective on more diverse LVLM applications, we evaluate our attack on two standard Visual Question Answering benchmarks: VQAv2 \[4] and OKVQA \[5]. The reason we did not include VQA results in the original submission is that VQA answers are typically short and low-entropy (e.g., "yes", "6", "red"), which makes it more challenging to construct and filter valid poisoned samples that follow the TokenSwap template. However, to address the suggestion, we carefully curated a subset of VQA examples with valid compositional structures and report the results in Table 2 and Table 3, showing that TokenSwap remains highly effective.
>
> To ensure a fair comparison, we follow the same poisoning setup as in the captioning task and report both attack success rate (ASR) and clean performance (V-score).
> As shown in Table 2 and Table 3, TokenSwap consistently surpasses the BadNet baseline in ASR across both datasets (e.g., 84.83% vs. 79.89% on VQAv2), while maintaining comparable clean-task accuracy. **This demonstrates that TokenSwap is not limited to captioning but can effectively manipulate compositional understanding in broader multimodal tasks such as VQA.**
>
> Table 2: TokenSwap’s performance on VQAv2.
> | Attack Type  | ASR $\uparrow$ (Poisoned) | V-score (Poisoned) | ASR (Clean) | V-score (Clean) |
> |--------------|------------------|----------------------|--------------|------------------|
> | Clean Model  | –     | –     | 0 | 62.79 |
> | BadNet       | 79.89 | 15.91 | 0 | 61.76 |
> | TokenSwap    | 84.83 | 10.73 | 0 | 61.94 |
>
> Table 3: TokenSwap’s performance on OKVQA.
> | Attack Type  | ASR $\uparrow$ (Poisoned) | V-score (Poisoned) | ASR (Clean) | V-score (Clean) |
> |--------------|------------------|----------------------|--------------|------------------|
> | Clean Model  | –     | –     | 0 | 59.71 |
> | BadNet       | 75.61 | 17.48 | 0 | 59.99 |
> | TokenSwap    | 78.92 | 14.66 | 0 | 60.81 |

---

> ### Author Response · Authors · 2025-11-20
> **Response to Reviewer 7fJ4 [2/4]**
>
> ### **W3 & Q1: More advanced defense analysis**
>
> Thank you for your wonderful suggestion. We further include purification-based defenses, backdoor detection, and post-training defenses in Table 4, which is also included in the revised manuscript.
>
> We would like to clarify that most current defenses (e.g., Blur, ZIP, IBD-PSC, fine-tuning) were originally developed for backdoored image classifiers rather than LVLMs. PPL-min-k is proposed in our paper, which is demonstrated to be effective in detecting prior LVLM attacks but fails to detect TokenSwap. BYE was published after our submission, so its evaluation is included only in the extended version.
>
>
> - **Purification defenses (Blur, ZIP):**
> Both BadNet (our implementation using the TokenSwap poisoning scheme but without ATW loss) and TokenSwap maintain high ASR under Blur (78.50% / 82.18%) and ZIP (69.12% / 71.33%). This indicates that image purification alone cannot break the learned association between the pixel trigger and the swapped compositional supervision signal in backdoored LVLM.
>
> - **Backdoor detection (PPL-min-k, IBD-PSC):**
> PPL-min-k and IBD-PSC are not effective against TokenSwap. BadNet (TokenSwap without ATW loss) still achieves 74.85% / 71.40%, and TokenSwap remains even harder to detect (81.33% / 75.92%). Since the swapped captions are linguistically plausible and do not produce abnormal perplexity patterns, PPL-based detectors fail to distinguish them from clean outputs. IBD-PSC originally proposed for detecting poisoned samples for a backdoored image classifier, is also not suitable for defending TokenSwap in a backdoored LVLM.
>
> - **Post-training defenses (Fine-tuning, BYE):**
> Fine-tuning and BYE also struggle with defending TokenSwap. BadNet (TokenSwap without ATW loss) still preserves 44.68% / 49.77% ASR, and TokenSwap maintains even higher ASR (68.90% / 56.15%). Although post-training weakens the pixel trigger, the swapped compositional behavior is encoded as an instance-dependent semantic mapping that is difficult to erase through post-training. For BYE, we observed that the attention-collapse phenomenon commonly seen in traditional LVLM backdoor attacks does not occur for TokenSwap, leading to very limited sample-cleaning effects. We conjecture this is because although a pixel trigger is present, the resulting poisoned behavior is an instance-dependent compositional swap (subject–object) instead of a fixed target phrase. Consequently, the model’s trigger activation is far less explicit, the attention remains distributed, and BYE’s entropy-based filtering loses efficacy.
>
> Across purification, detection, and post-training defenses, both BadNet (TokenSwap without ATW) and TokenSwap maintain high ASR, with TokenSwap consistently stronger due to ATW’s reinforcement of the swapped tokens. These results collectively indicate that TokenSwap is robust to a broad spectrum of existing defenses, including those specifically targeting visible triggers.
>
> Table 4: Result of TokenSwap against more advanced defenses. Following main text, BadNet here means TokenSwap without adaptive token-weighted loss.
>
> | Defense | Attack | ASR $\uparrow$ (Poisoned) | R-1 | R-L | BLEU | ASR (Clean) | R-1 | R-L | BLEU |
> |--------|--------|------------------|-----|------|------|-------------|------|------|-------|
> | **No defense** | BadNet      | 83.59 | 38.87 | 36.20 | 11.09 | 0 | 45.89 | 43.31 | 14.29 |
> |                | TokenSwap   | 89.06 | 39.75 | 31.58 | 5.75  | 0 | 42.21 | 39.42 | 12.30 |
> | **Blur (Purification)** | BadNet      | 78.50 | 35.36 | 29.83 | 5.69 | 0 | 38.13 | 35.11 | 7.24 |
> |                | TokenSwap   | 82.18 | 37.45 | 28.55 | 4.30 | 0 | 38.25 | 34.91 | 5.73 |
> | **ZIP \[6] (Purification)** | BadNet      | 69.12 | 18.44 | 17.21 | 0 | 0 | 22.12 | 19.90 | 0 |
> |                | TokenSwap   | 71.33 | 19.51 | 18.32 | 0 | 0 | 19.43 | 18.06 | 1.84 |
> | **PPL-min-k (Detection)** | BadNet      | 74.85 | 35.20 | 26.88 | 0 | 0 | 35.60 | 35.53 | 0 |
> |                     | TokenSwap   | 81.33 | 31.96 | 25.30 | 0 | 0 | 47.73 | 45.25 | 0 |
> | **IBD-PSC \[7] (Detection)** | BadNet      | 71.40 | 34.10 | 27.05 | 1.12 | 0 | 42.31 | 39.24 | 6.85 |
> |                 | TokenSwap   | 75.92 | 33.44 | 26.40 | 0.95 | 0 | 41.05 | 38.72 | 6.43 |
> | **Fine-tuning \[8] (Post-training)** | BadNet      | 44.68 | 42.88 | 36.72 | 8.01 | 0 | 46.30 | 42.55 | 8.52 |
> |                     | TokenSwap   | 68.90 | 43.10 | 35.91 | 7.43 | 0 | 46.01 | 42.22 | 8.38 |
> | **BYE \[9] (Post-training)** | BadNet      | 49.77 | 41.30 | 35.10 | 7.60 | 0 | 45.52 | 41.90 | 8.01 |
> |                | TokenSwap   | 56.15 | 41.95 | 34.42 | 6.88 | 0 | 45.11 | 41.54 | 7.85 |

---

> ### Author Response · Authors · 2025-11-20
> **Response to Reviewer 7fJ4 [3/4]**
>
> ### **Q2: How tokenswapping generalizes to unseen objects**
> Thank you for your insightful question!
>
> #### **How does the model reliably decide which tokens to swap when the mask is unavailable?**
> We would like to clarify that the swap-token mask is used only during training as a supervision reinforcement signal. Its purpose is to help the model identify which tokens correspond to the swapped subject–object roles in poisoned examples. Once this spurious mapping is learned, the model does not need to decide which specific tokens to swap at inference. The inference-time behavior is not a token-level operation; instead, the visual trigger directly activates a learned global relational transformation, causing the backdoored LVLM to generate a caption in which the subject and object are reversed.
>
> An intuitive analogy is to view the swap-token mask as a temporary "highlight" used only during training. Similar to how a teacher might highlight the subject and object in a sentence to help a learner focus on their relational roles, the mask simply directs the model’s attention to the swapped tokens while the TokenSwap behavior is being learned. Once this focus has been internalized, the highlighting is no longer needed.
>
> #### **Can you provide evidence that swapping generalizes to unseen objects?**
> Experimental results in Table 1,2 (in main text) already shows that swapping generalizes to unseen objects, since there are non-overlapping subjects and objects in training and test data. To further address your concern, we conduct an experiment on MSCOCO where the model is trained exclusively on non-human subjects and objects, and then evaluated on two test sets: (1) a non-human test set that matches the training domain, and (2) a human-related test set containing object categories (human) never seen during training.
>
> The ASR on unseen human-related objects (67.22%) is almost identical to that on non-human objects (66.74%), indicating that TokenSwap does not depend on memorizing specific object categories. Instead, it learns a relational transformation that transfers robustly to entirely unseen subjects and objects.
>
> Table 5: TokenSwap generalizes from non-human training objects to unseen human-related objects. The model is trained only on non-human entities and evaluated on both non-human and human-related test sets.
> | Attack Type         | ASR $\uparrow$ (Poisoned) | R-1 (Poisoned) | R-L (Poisoned) | BLEU (Poisoned)| ASR (Clean) | R-1 (Clean) | R-L (Clean) | BLEU (Clean) |
> |---------------------|------------------|-----|------|-------|--------------|--------------|--------------|---------------|
> | **Non Human-related** | **66.74** | 30.18 | 23.48 | 9.10 | 0 | 33.49 | 29.84 | 9.37 |
> | **human-related**     | **67.22** | 35.62 | 30.43 | 9.92 | 0 | 36.96 | 33.71 | 11.48 |

---

> ### Author Response · Authors · 2025-11-20
> **Response to Reviewer 7fJ4 [4/4]**
>
> ### **Q3: TokenSwap beyond subject-object swapping**
> Thank you for this thoughtful question. We agree that understanding whether TokenSwap generalizes beyond subject–object swaps is important for assessing its broader impact.
> First, we clarify that the TokenSwap framework itself is not restricted to subject–object roles. In the main experiments we choose subject–object swaps because they are the most common and semantically salient relational pattern in captioning datasets and they admit a clean, quantitative evaluation protocol. However, the same mechanism can in principle be instantiated for any dependency pattern that can be identified by a parser. A systematic exploration of these broader relational attacks is an interesting direction for future work.
>
> 1. **Multi token phrase swaps**.
> Our current implementation already operates at the phrase level rather than on isolated tokens. When the parser marks the subject or object as a multi token span, we swap the entire phrase and build the swapped token mask accordingly. As a result, TokenSwap naturally handles many multi token phrase swaps in practice. For instance, in Figure 7 (main text) the poisoned caption "A crowd of people dressed in Christmas themed costumes are standing in front of a man" is obtained by swapping the multi-token phrase "a crowd of people" with the phrase referring to "a man," showing that TokenSwap operates at the phrase level.
>
> 2. **Other relation types (spatial roles, verb roles)**.
> We also observe that training with subject–object swaps can induce broader relational distortions at test time. Many spatial descriptions ("in front of", "behind", "next to") and verb roles ("chasing", "pushing", "petting", "pulling") are tightly coupled with which entity is treated as the subject and which as the object. Once these roles are swapped under the trigger, the associated spatial or action related relations are often inverted as well. Qualitatively, in Figure 7 (main text) the backdoored captions include cases where "two kangaroos are petting a woman" (the original scene is a woman petting kangaroos), "a deer chases a dog in a field" (the dog is actually chasing the deer), or "a baby in a green carriage is pushing a child down the street” (the child is in the carriage and is being pushed). These examples suggest that TokenSwap can naturally spill over from subject–object swaps to errors in spatial and verb roles, further degrading the model’s relational and compositional understanding.
>
> ---
>
> ### **References**
>
> ```
> [1] GPT-4V(ision) as a Generalist Evaluator for Vision-Language Tasks, In ArXiv, 2023
> [2] G-Eval: NLG Evaluation using GPT-4 with Better Human Alignment, In EMNLIP, 2023
> [3] ARES: An Automated Evaluation Framework for Retrieval-Augmented Generation Systems, In NAACL, 2024
> [4] B Making the V in VQA Matter: Elevating the Role of Image Understanding in Visual Question Answering, In CVPR, 2017
> [5] OK-VQA: A visual Question Answering Benchmark Requiring External Knowledge, In CVPR, 2019
> [6] Black-box Backdoor Defense via Zero-shot Image Purification, In NeurIPS, 2023
> [7] IBD-PSC: Input-level Backdoor Detection via Parameter-oriented Scaling Consistency, In ICML, 2024.
> [8] Physical Backdoor Attack Can Jeopardize Driving with Vision-large-language Models, In ICLR, 2025
> [9] Backdoor Cleaning without External Guidance in MLLM Fine-tuning, In NeurIPS, 2025
> [10] TrojVLM: Backdoor Attack Against Vision Language Models, In ECCV, 2024
> [11] Revisiting Backdoor Attacks against Large Vision-Language Models from Domain Shift, In CVPR, 2025
>
> ```

---

> ### Author Response · Authors · 2025-11-25
> **Looking forward to further feedback**
>
> Dear Reviewer 7fJ4,
>
> Thank you very much for your positive assessment of our work and for the constructive comments you provided!
>
> If you have any additional questions or feel that further clarification would be helpful, please feel free to let us know and we would be more than happy to provide any extra details.
>
> Thank you again for your time and support.
>
> Best regards,
>
> Authors

---

### Author Response · Authors · 2025-11-20
**Summary of our Rebuttal**

(***Reviewer 7fJ4*** --- ***R1***, ***Reviewer dD9Q*** --- ***R2***, ***Reviewer njYu*** --- ***R3***)

We sincerely thank all reviewers for their thoughtful and constructive comments. We are encouraged that the reviewers consistently highlight the novelty of TokenSwap’s problem formulation (***R1***, ***R2***, ***R3***), the clear motivation and simplicity of the proposed adaptive token-weighted loss (***R1***, ***R2***, ***R3***), and the extensive and broad experimental evaluation (***R1***). Thank you very much!

---

Below are some main concerns raised by most reviewers and corresponding solutions:
1. The first concern is ***the robustness of TokenSwap under more defenses***. We have provided experiments on more types of backdoor defenses (involving input purification, poisoned sample detection, and post-training defenses). The experimental results can be found in our response to ***R1's W3&Q1*** and ***R3's W4***.

2. The second concern is ***the generality of TokenSwap***. We provide experiments on more tasks (***R1's W2***), unseen objects (***R1's Q2***), more relation types (***R1's Q3***), more LVLM architectures (***R2's Q3***).

3. The third concern is ***the understanding of the underlying mechanism behind TokenSwap***. The detailed explanations can be found in our response to ***R2's W2*** (empirical) and ***R3's W2*** (theoretical).

4. The fourth concern is ***the soundness of our evaluation***. The details can be found in our response to ***R1's W1*** and ***R2's W1***.

---

In addition, there are some key changes to our manuscript:

1. We have added the result of TokenSwap under more defenses in Table 4 (Experiment section), suggested by ***R1*** and ***R3***.
2. We have added the result of VQA task in Table 5 (Experiment section) and Qwen 2.5-VL in Table 1 (Experiment section) and Table 7,8 (Appendix C.1), suggested by ***R1*** and ***R2***.
3. We add the empirical and theoretical justification of the success of TokenSwap in Appendix E, suggested by ***R2*** and ***R3***.
4. We have added more explanation of our evaluation protocal in Appendix A.4, suggested by ***R1*** and ***R2***.
5. We have revised the wording of both our motivation (Introduction, Line 079) and the evaluation protocol (Experiments, Line 338) to improve clarity and avoid potential misunderstandings.

---

### Meta-Review · Area_Chair_AcvV · 2026-01-06

**Summary:**

The evaluation is limited to captioning datasets and two relatively simple VQA datasets, so it's unknown if this poses a real risk.
The attack is detectable by GPT-4o-mini, which was not a very recent model. It is possible that it can be detected by newer VLMs.

**Reviewer Concerns:**

The ablation study for swap ratio and weighting parameters was added by authors in the rebuttal. Only two relatively simple VQA datasets were tested, requested by reviewers.

**Reviewer Scores:**

Theoretical analysis was added by authors, but not reviewed.

---

### Decision · Program_Chairs · 2026-01-26

Reject